# Source Apportionment and Health Risk Assessment of Heavy Metals in Eastern Guangdong Municipal Solid Waste

**Zhihua Tang** [1,2,*], **Minru Liu** [1,2], **Linzi Yi** [1,2], **Huafang Guo** [1,2], **Tingping Ouyang** [3], **Hua Yin** [1,2] **and Mingkun Li** [3]

1   Key Laboratory of Renewable Energy, Guangzhou Institute of Energy Conversion, Chinese Academy of Sciences, Guangzhou 510640, China; liumr@ms.giec.ac.cn (M.L.); yilz1@ms.giec.ac.cn (L.Y.); guohf@ms.giec.ac.cn (H.G.); yinhua@ms.giec.ac.cn (H.Y.)
2   Guangdong Key Laboratory of New and Renewable Energy Research and Development, Guangzhou 510640, China
3   School of Geography, South China Normal University, Guangzhou 510631, China; oyangtp@m.scnu.edu.cn (T.O.); limk@m.scnu.edu.cn (M.L.)
*   Correspondence: tangzh@ms.giec.ac.cn; Tel./Fax: +86-020-3845-5987

**Abstract:** This research focused on the contents of the five most bio-toxic heavy metals, As, Cd, Hg, Cr, and Pb of 26 municipal solid waste (MSW) samples from the Eastern Guangdong Area. To investigate the apportion of the heavy metal source, Pearson correlation and principal component analysis (PCA) were introduced as major approaches. The health risks posed to MSW workers exposed to heavy metals in MSW were assessed using a Monte Carlo simulation combined with the US Environmental Protection Agency Health Risk Assessment Model. The As, Cd, Hg, Cr, and Pb contents of the east Guangdong MSW were (0.76 ± 0.75), (2.14 ± 4.44), (0.11 ± 0.14), (55.42 ± 31.88), and (30.67 ± 20.58) mg/kg, respectively. Hg, Cr, and Pb were potentially derived from glass, textile, food waste, and white plastic, while As and Cd were mainly derived from soil and food waste in the MSW. The non-carcinogenic risks of heavy metal in MSW exposure to MSW workers could be ignored. However, the heavy metals in MSW might pose carcinogenic risks, with the probabilities for male and female workers being 35% and 45%, respectively. The non-carcinogenic and carcinogenic risk indices were slightly higher for female workers under the same exposure situations.

**Keywords:** MSW; heavy metals; source apportionment; MSW workers; health risk

## 1. Introduction

The net volume of municipal solid waste (MSW) in China is 215 million tons per year with an annual growth rate of 3~5%, and the number of MSW workers is over 4 million [1]. MSW contains larger amounts of heavy metals, seriously endangering human health [2–5]. In many cities of China, including Guangzhou, Beijing, Shanghai, Dalian, Chengdu and Nanchang, the mean As, Cd, Hg, Cr, and Pb concentrations in MSW were found to exceed their corresponding threshold values of the national standard [6–8]. MSW workers are the main force of waste collection, sorting, transportation, and disposal of waste. This leads to serious considerations regarding potential health risk of MSW workers due to long working time and high frequency exposed to MSW heavy metals. It is of great significance to discuss the possible sources of heavy metals in MSW and quantitatively evaluate their effects on the health of MSW workers.

The source apportionment of MSW heavy metals and its health risk assessment have become a hot topic of academic research around the world [4,9–15]. Previous studies found that components of

metal, metal-coated materials, food waste, soil, plastic, and paper were the main heavy metal sources of MSW [9]. Many health risk studies mainly focused on the impact of toxic heavy metals in the emissions from MSW incinerators on nearby citizens [3,4,13]. However, relatively little research has been done on the health risks of MSW workers who are directly exposed to MSW heavy metals.

MSW workers are a particular population that has made great contributions to urban living environment. However, due to lack of self-protection awareness due to high illiteracy, many of them suffered from different types of occupational disease like headaches, nausea, allergies and cancers [11,15,16]. The occupational risks mainly derived from different exposure pathways for a variety of toxic pollutants, such as toxic heavy metals through hand-mouth ingestion, dermal absorption, and air inhalation [17]. The main purpose of this study were: (1) to determine the sources of the heavy metals in the Eastern Guangdong MSW using the Pearson correlation and principal component analysis (PCA) methods; (2) to assess health risks caused by heavy metals posed to MSW workers using a Monte Carlo simulation combined with the US Environmental Protection Agency (US EPA) health risk assessment model. The research results could provide a scientific basis for MSW management and MSW workers' health protection in the Eastern Guangdong.

## 2. Materials and Methods

### 2.1. Study Area and Sample Collection

The Eastern Guangdong, ranging from longitude of 115° E to 117° E and latitude of 22° N to 25° N, is comprised of five cities (Shantou, Shanwei, Jieyang, Chaozhou, and Meizhou) (refer to Figure 1). It has a permanent population of 18 million with 27,000 tons of MSW generated per day. The harmless disposal rate of MSW in east Guangdong reached 93% [18], with sanitary landfill as the major disposal method. According to the "Methods of Sampling and Physical Analysis of Municipal Solid Waste (CJ/T 313-2009)" published by Ministry of Housing and Urban-Rural Development of the People's Republic of China [19], a total of 26 samples were collected from the main waste compression stations, transfer stations and landfills. The sampling sites were denoted as black dots in Figure 1. In each sampling site, the trapezoidal body was first formed by the MSW dump, and then the samples were collected from all sides of the trapezoidal body. Furthermore, in order to make the sample more representative, MSW were collected on a continued basis on different days in one week, namely, every weekday at each sampling site. Subsequently, all sub-samples obtained from each sampling site were completely mixed manually prior to being filled in air-tight plastic bags. Finally, 50 kg of MSW was collected from each sampling site.

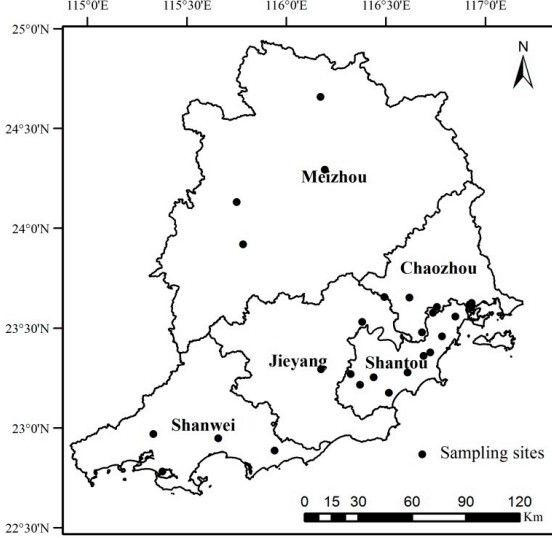

**Figure 1.** Study area map with location of MSW sampling sites.

*2.2. Sample Preparation and Testing*

All samples were processed within 24 h of collection. Firstly, the visible and separable components were roughly sorted in each MSW sample manually and the remaining sample was sieved through to a mesh with the pore size of 10 mm. The screen overflow was sorted to corresponding components and the underflow was sorted according to its components. Subsequently, the sorted samples were placed in separated containers and then moved to the electric thermostatic-drying oven for drying at $(105 \pm 5)$ °C until a constant weight was reached. Each component was weighed on a balance (0.001 kg accuracy), so their mass fractions can be calculated. After weighing, a crasher was used to grind the samples to the diameter of less than 0.5 mm. After adequate mixing, the quartering method was used on a continued basis to contract and split the sample until the mass was reduced to approximately 100 g and then stored in dried jars at room temperature. When the heavy metal test was conducted, 2 g of sample was extracted and the grinder was used to pulverize the particle to a diameter of less than 0.5 mm. For each specimen, 1 g was taken for testing.

The typical toxic heavy metals (As, Cd, Hg, Cr, and Pb) were selected to carry out the test. For As, Cd, Cr, and Pb to be tested, the dry samples were dissolved at $(95 \pm 5)$ °C by 10 mL of 1:1 $HNO_3$, 5 mL of 65% $HNO_3$, 3 mL of 30% $H_2O_2$ and 10 mL of HCl. Then, the inductively coupled plasma optical emission spectrometry method (ICP-OES) were used to conduct the test (Instrument Model PerkinElmer Optima 8000DV). For Hg to be tested, the samples were digested using $HNO_3$, HCl, $KMnO_4$ and $K_2S_2O_8$ and cold atomic absorption spectrophotometry was applied for testing (Instrument Model Leemanlabs Hydra II AF Gold). To ensure quality, heavy metals were tested to such national standards as GB 5085.3-2007 Appendix A and CJ/T 98-1999 published by State Environmental Protection Administration and Ministry of construction of the People's Republic of China, respectively [20,21]. The container used in the experiment was first washed by 10% hot nitric acid and then rinsed by de-ionized water to achieve a blank background. The instrument was pre-heated 1 h prior to use in order to avoid wavelength drift. Moreover, the samples were tested in different batches, and two blank samples and reference samples were added for every ten groups of sample test to examine the degree of wavelength drift. To ensure precision and accuracy, each sample was assigned with four parallel samples, and the mean values of results were indicated by the final concentration of heavy metals.

*2.3. Statistical Analysis*

In order to determine composition of MSW and distribution of heavy metal contents in the MSW, the descriptive statistical method was used to analyze the mass percentage and heavy metal concentrations in the dry basis. Considering the widespread application of Pearson Correlation to the heavy metal source apportionment [22–24], Pearson Correlation we used to explore the correlation between heavy metal contents and their mass percentage in the components of municipal solid waste. Principal component analysis was one of commonly used method of multivariate statistical analysis to identify the factors that could affect heavy metal contents and the potential sources of heavy metals to MSW [22–24]. In order for an in-depth investigation into the potential sources of As, Cr, Hg, Cd, and Pb in the Eastern Guangdong MSW, after Pearson correlation analysis, the principal component analysis was conducted to extract the main independent factors from the large set of variables by reducing the dimensionality of the dataset. All the above statistical analyses were completed with the assistance of SPSS 19.0.

*2.4. Human Health Risk Assessment*

2.4.1. Health Risk Assessment Model

The Dose-Response Model recommended by the US EPA was conducted to determine the health risks posed by MSW heavy metals. According to the model theory, the three primary heavy metals exposure pathways to MSW workers are hand-mouth ingestion, dermal absorption, and air inhalation. The average daily intake (ADI, mg/kg/day) of heavy metal via the three exposure pathways can be

calculated using Equations (1)–(3) [25]. Hazard quotient (HQ) of non-carcinogens can be determined subsequently by dividing the ADI of each exposure pathway by the corresponding reference dose (RfD, mg/kg-day) [25]. Furthermore, the hazard index (HI) is defined by adding HQs of each non-carcinogens or each exposure pathway together to assess the combination non-carcinogenic risks posed by multiple non-carcinogens and/or more than one exposure pathways (refer to Equation (4)) [25]. Thus, the total exposure hazard index (TEHI) is the sum of HIs to estimate the overall potential non-carcinogenic hazard posed by all of non-carcinogens to the MSW workers through all exposure pathways (refer to Equation (5)) [25]. Whereas, the level of carcinogenic risk caused by carcinogens is calculated through multiplying the ADI by the corresponding slope factor (SF, per mg/kg-day). Similarly, the total carcinogenic risk index (TCRI) is the sum of carcinogenic risk of each carcinogen across all exposure pathways (refer to Equation (6)) [25].

$$\text{ADI}_{\text{ing}} = C_{\text{MSW}} \times \frac{\text{IngR} \times \text{EF} \times \text{ED}}{\text{BW} \times \text{AT}} \times 10^{-6} \tag{1}$$

$$\text{ADI}_{\text{dermal}} = C_{\text{MSW}} \times \frac{\text{SA} \times \text{AF} \times \text{ABS} \times \text{EF} \times \text{ED}}{\text{BW} \times \text{AT}} \times 10^{-6} \tag{2}$$

$$\text{ADI}_{\text{inh}} = C_{\text{MSW}} \times \frac{\text{InhR} \times \text{EF} \times \text{ED}}{\text{PEF} \times \text{BW} \times \text{AT}} \tag{3}$$

$$\text{HI} = \sum \text{HQ}_i = \sum \frac{\text{ADI}_i}{\text{RfD}_i} \tag{4}$$

$$\text{TEHI} = \sum \text{HI} = \sum_{j=1}^{N} \frac{\text{ADI}_{\text{ing}}^{j}}{\text{RfD}_{\text{ing}}^{j}} + \sum_{j=1}^{N} \frac{\text{ADI}_{\text{dermal}}^{j}}{\text{RfD}_{\text{dermal}}^{j}} + \sum_{j=1}^{N} \frac{\text{ADI}_{\text{inh}}^{j}}{\text{RfD}_{\text{inh}}^{j}} \tag{5}$$

$$\text{TCRI} = \sum_{j=1}^{N} \left( \text{ADI}_{\text{ing}}^{j} \times \text{SF}_{\text{ing}}^{j} \right) + \sum_{j=1}^{N} \left( \text{ADI}_{\text{dermal}}^{j} \times \text{SF}_{\text{dermal}}^{j} \right) + \sum_{j=1}^{N} \left( \text{ADI}_{\text{inh}}^{j} \times \text{SF}_{\text{inh}}^{j} \right) \tag{6}$$

where $\text{ADI}_{\text{ing}}$, $\text{ADI}_{\text{dermal}}$, and $\text{ADI}_{\text{inh}}$ are the average daily intake from hand-mouth ingestion, dermal absorption and air inhalation, respectively (mg/kg·day); $C_{\text{MSW}}$ is the heavy metal concentration of MSW (mg/kg); IngR is the rate of hand-mouth intake (mg/day); InhR is the rate of breath inhalation (m$^3$/day); EF is the exposure frequency (day/year); ED is the exposure duration (year); BW is the body weight of the exposed individual (kg); AT is the time period over which the dose is averaged (day); PEF is the emission factor(kg/m$^3$); SA is the exposed skin surface area (cm$^2$); AF is the adherence factor (kg/cm$^2$·day); ABS is the dermal absorption factor (unitless); RfD is a reference dose (mg/(kg·day)); HQ is the non-carcinogenic risk quotient; SF is the carcinogenic slope factor; HI is the non-carcinogenic risk index; TEHI is the total non-carcinogenic index; TCRI is the carcinogenic risk index; *i* indicates the number of heavy metals or exposure pathways; N is the number of heavy metals.

### 2.4.2. Monte Carlo Simulation

The Monte Carlo simulation is widely applied in risk assessments [26–28]. Since the MSW sources were complicated and the heavy metal concentrations of MSW varied among seasons [29], complexity and uncertainty inherently arise for health risk assessment [30]. To scientifically evaluate the health risk of MSW heavy metals, the uncertain parameters should apply random distribution functions instead of constants and the risk assessment model should be solved by Monte Carlo method until the numerical results are convergent or reached expected confidential interval of probability. Thus, the results gained from the Monte Carlo simulation were the probability distribution of the health risks, which were more reliable than the method of using fixed parameter values as input.

The distribution functions of $C_{\text{MSW}}$ were obtained from testing data. Generally, the longest working life of MSW workers is approximately 30 years. To consider the highest risk of heavy metals exposure to human body, the ED value should be taken 30 years. Furthermore, considering the

physiological and behavioral differences, MSW workers were divided into male workers and female workers. Parameters' distribution taken from the references of other related studies were listed in Table A1 [31–36]. The values of RfD and SF refer to previous studies are listed in Table A2 [31,32].

## 3. Results

### 3.1. Components of Eastern Guangdong MSW

The statistics of the mass fractions of MSW components in the dry basis were illustrated in Figure 2 as box plot. The result showed that the MSW was majorly constituted by soil, glass, metal, paper, plastic, textile, grass, food waste, and white plastic. The order of mass fraction of each component in the dry basis was food waste > plastic > soil > paper > glass > grass > textile > white plastic > metal. Among all of the components, food waste accounts for 13.71~35.09%, with an average of 24.64%; plastic accounts for 10.53~28.90%, with an average of 17.95%; soil accounts for 8.86~32.32%, with an average of 16.98%; paper accounts for 4.83~26.78%, with an average of 13.99%; glass accounts for 0~20.87%, with an average of 8.06%; grass accounts for 0.90%~14.77%, with an average of 6.60%; textile accounts for 0~14.04%, with an average of 5.58%; white plastic accounts for 0.33~9.95%, with an average of 3.93%; metal accounts for 0~8.22%, with an average of 2.26%.

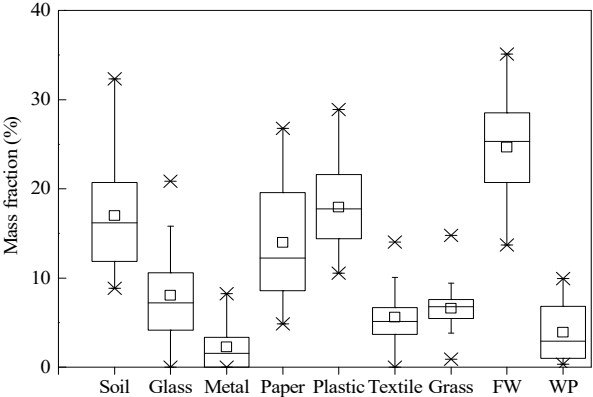

**Figure 2.** Mass fraction of MSW components on a dry weight basis (FW: food waste, WP: white plastic).

### 3.2. Heavy Metal Concentrations in MSW

The As, Cd, Hg, Cr and Pb concentrations in the Eastern Guangdong MSW were (0.76 ± 0.75), (2.14 ± 4.44), (0.11 ± 0.14), (55.42 ± 31.88) and (30.67 ± 20.58) mg/kg, respectively (refer to Table 1).

**Table 1.** Statistical results of heavy metal contents in 26 MSW samples (mg/kg).

| Statistical Parameter | Elements | | | | |
|---|---|---|---|---|---|
| | As | Cd | Hg | Cr | Pb |
| Minimum | 0.08 | – | – | 10.71 | 2.61 |
| Maximum | 2.01 | 20.21 | 0.54 | 118.98 | 80.15 |
| Mean | 0.76 | 2.14 | 0.11 | 55.42 | 30.67 |
| Standard deviation | 0.75 | 4.44 | 0.14 | 31.88 | 20.58 |
| variable coefficient | 99% | 207% | 127% | 58% | 67% |

Note: – means no detection; The detection limits of As, Cd and Hg are 0.03, 0.002 and 0.05 µg/mL, respectively.

### 3.3. Pearson Correlation and Principal Component Analysis

The result of Pearson correlation analysis between heavy metal contents and MSW components' mass fraction is shown in Table 2.

**Table 2.** Pearson correlation coefficients between heavy metals and MSW components' mass fraction.

| | As | Cd | Hg | Cr | Pb | Soil | Glass | Metal | Paper | Plastic | Textile | Grass | FW | WP |
|---|---|---|---|---|---|---|---|---|---|---|---|---|---|---|
| As | 1 | | | | | | | | | | | | | |
| Cd | 0.45 | 1 | | | | | | | | | | | | |
| Hg | −0.80 ** | −0.29 | 1 | | | | | | | | | | | |
| Cr | −0.85 ** | −0.26 | 0.19 | 1 | | | | | | | | | | |
| Pb | −0.75** | 0.2 | 0.08 | 0.62 ** | 1 | | | | | | | | | |
| Soil | 0.89 ** | 0.15 | −0.50 * | −0.43 * | −0.61 ** | 1 | | | | | | | | |
| Glass | −0.52 | 0.17 | −0.07 | 0.29 | 0.57 ** | −0.37 | 1 | | | | | | | |
| Metal | 0.56 | −0.28 | −0.15 | −0.16 | −0.31 | −0.04 | −0.50 ** | 1 | | | | | | |
| Paper | −0.32 | −0.50 * | 0.23 | −0.15 | −0.52 ** | 0.21 | −0.67 ** | 0.3 | 1 | | | | | |
| Plastic | 0.39 | −0.35 | −0.14 | −0.09 | −0.40 * | −0.06 | −0.57 ** | 0.39 | 0.50 ** | 1 | | | | |
| Textile | 0.24 | 0.29 | −0.03 | 0.08 | 0.45 * | −0.48 * | 0.11 | 0.04 | −0.45 * | −0.03 | 1 | | | |
| Grass | −0.27 | −0.16 | 0.34 | 0.38 | 0.42 * | −0.58 ** | 0.06 | −0.15 | −0.03 | 0.13 | 0.33 | 1 | | |
| FW | −0.89 ** | 0.48 * | 0.16 | 0.34 | 0.64 ** | −0.63 ** | 0.53 ** | −0.28 | −0.58 ** | −0.46 * | 0.25 | 0.14 | 1 | |
| WP | −0.2 | 0.25 | 0.15 | 0.11 | 0.47 * | −0.16 | 0.64 ** | −0.33 | −0.79 ** | −0.78 ** | 0.25 | −0.04 | 0.57 ** | 1 |

Note: * was significantly correlated at the 0.05 level (bilateral); ** was significantly correlated at 0.01 level (bilateral); — means negative correlation; FW: food waste; WP: white plastic.

The principal component analysis loadings for the heavy metal contents of the MSW samples were shown in Table A3. Two principal components have eigenvalues >1.0 and accounted for 96.1% of the total variance. The rotated factor analysis component matrix is shown in Table 3.

**Table 3.** Factor loading matrix after orthogonal rotation of maximum variance.

| Elements | Principal Component | |
|:---:|:---:|:---:|
| | F1 | F2 |
| As | −0.84 | 0.49 |
| Cd | 0.02 | 0.99 |
| Hg | 0.98 | 0.01 |
| Cr | 0.98 | −0.08 |
| Pb | 0.96 | 0.13 |

Notes: Extraction method: Principal components analysis; Rotation method: Varimax with Kaiser normalization; Rotation converged in 10 iterations.

### 3.4. Health Risk Assessment

Five heavy metals (As, Cd, Hg, Cr, and Pb) were considered in the health risk assessment because of their strong biotoxicity to MSW workers. The US EPA model was conducted to assess the non-carcinogenic risk and carcinogenic risk of MSW workers exposed to heavy metals through three pathways: (1) hand-to-mouth ingestion, (2) dermal absorption, (3) air inhalation. The simulation was performed using Crystal Ball 11.0 software, and 10,000 iterations were performed. The probability and frequency distribution of the non-carcinogenic hazard indices for different exposure pathways and posed by different heavy metals are illustrated in Figures 3 and 4, respectively. The probability and frequency distribution of the carcinogenic risk indices for different exposure pathways and posed by different heavy metals are listed in Figures 5 and 6, respectively.

### 3.4.1. Non-Carcinogenic Risk

Because of the RfD values of As and Pb for air inhalation were missing (refer to Table A2), only two exposure pathways (hand-to-mouth ingestion, and dermal absorption) of As and Pb were taken into consideration for the no-carcinogenic risk assessment.

The results in Figure 3 show that: (1) the mean non-carcinogenic hazard indices of hand-to-mouth ingestion, dermal absorption and air inhalation for male workers were $1.08 \times 10^{-1}$, $1.02 \times 10^{-6}$ and $3.72 \times 10^{-4}$, respectively; (2) the mean non-carcinogenic hazard indices of hand-to-mouth ingestion, dermal absorption and air inhalation for female workers were $1.27 \times 10^{-1}$, $1.07 \times 10^{-6}$ and $4.36 \times 10^{-4}$, respectively; (3) the mean total exposure hazard indices for male and female workers were $1.09 \times 10^{-1}$ and $1.28 \times 10^{-1}$, respectively. The hazard indices distribution of different exposure pathways indicated that the non-carcinogenic hazard indices of hand-to-mouth ingestion, dermal absorption and air inhalation for male and female workers were distributed in the range of $10^{-2}$~$10^{-1}$, $10^{-8}$~$10^{-6}$, and $10^{-5}$~$10^{-3}$, respectively. The hazard indices of hand-to-mouth ingestion were 5~6, and 2~3 orders of magnitude higher than that of dermal absorption and air inhalation, respectively.

The results in Figure 4 show that: (1) the average non-carcinogenic hazard indices of As, Cd, Hg, Cr and Pb for male workers were $8.04 \times 10^{-3}$, $1.32 \times 10^{-2}$, $1.52 \times 10^{-3}$, $5.75 \times 10^{-2}$ and $2.84 \times 10^{-2}$, respectively; (2) the average non-carcinogenic hazard indices of As, Cd, Hg, Cr and Pb for female workers were $9.43 \times 10^{-3}$, $1.55 \times 10^{-2}$, $1.78 \times 10^{-3}$, $6.75 \times 10^{-2}$ and $3.34 \times 10^{-2}$, respectively. The hazard indices distribution of different heavy metals suggested that the non-carcinogenic hazard indices of As, Cd, Hg, Cr and Pb for male and female workers were distributed in the range of $10^{-4}$~$10^{-2}$, $10^{-4}$~$10^{-2}$, $10^{-5}$~$10^{-3}$, $10^{-3}$~$10^{-1}$, and $10^{-4}$~$10^{-2}$, respectively. Therefore, Cr contributed the most to the non-carcinogenic hazard for male and female workers, following by As, Cd, Pb and Hg.

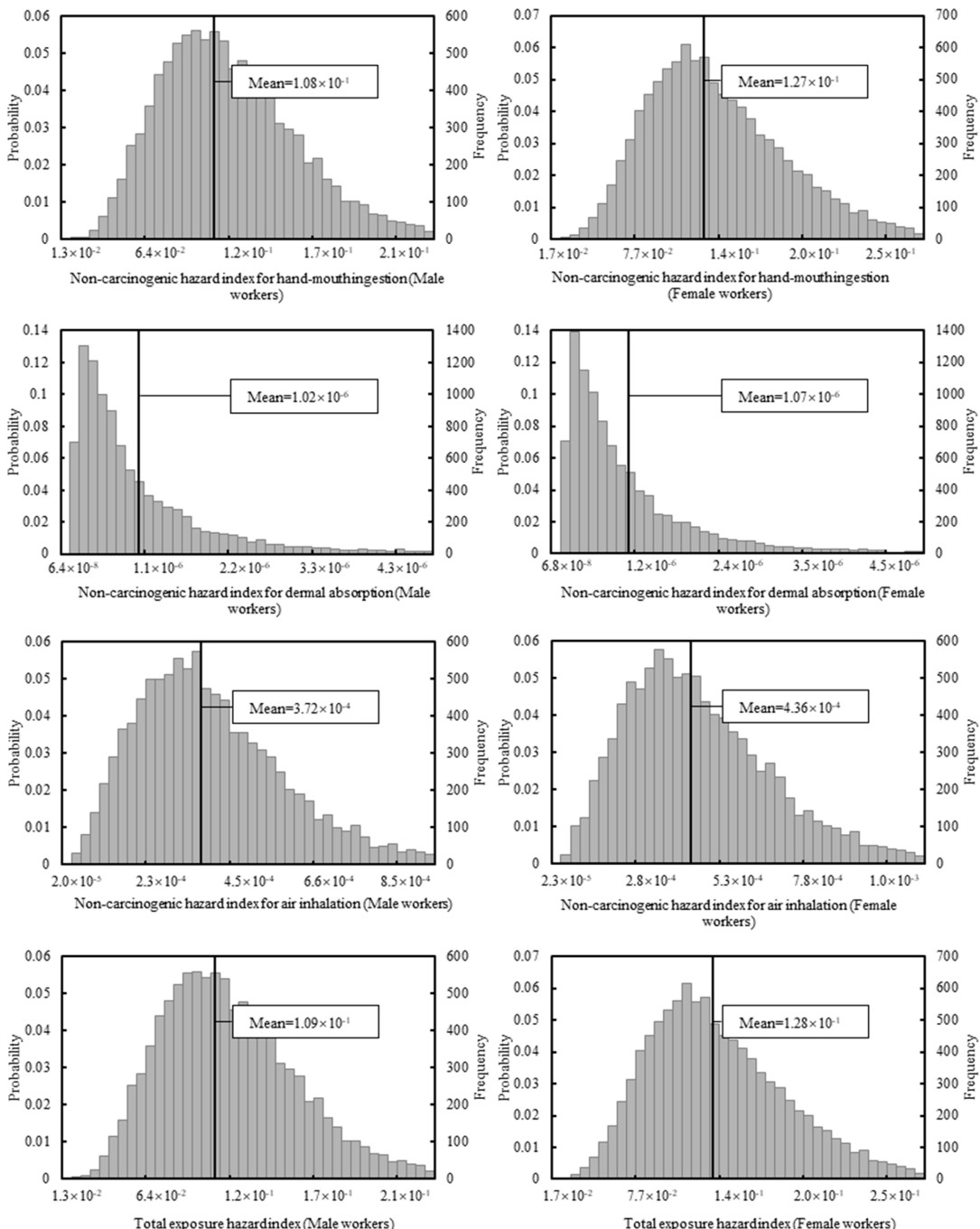

**Figure 3.** Distribution of non-carcinogenic risks posed through different exposure pathways.

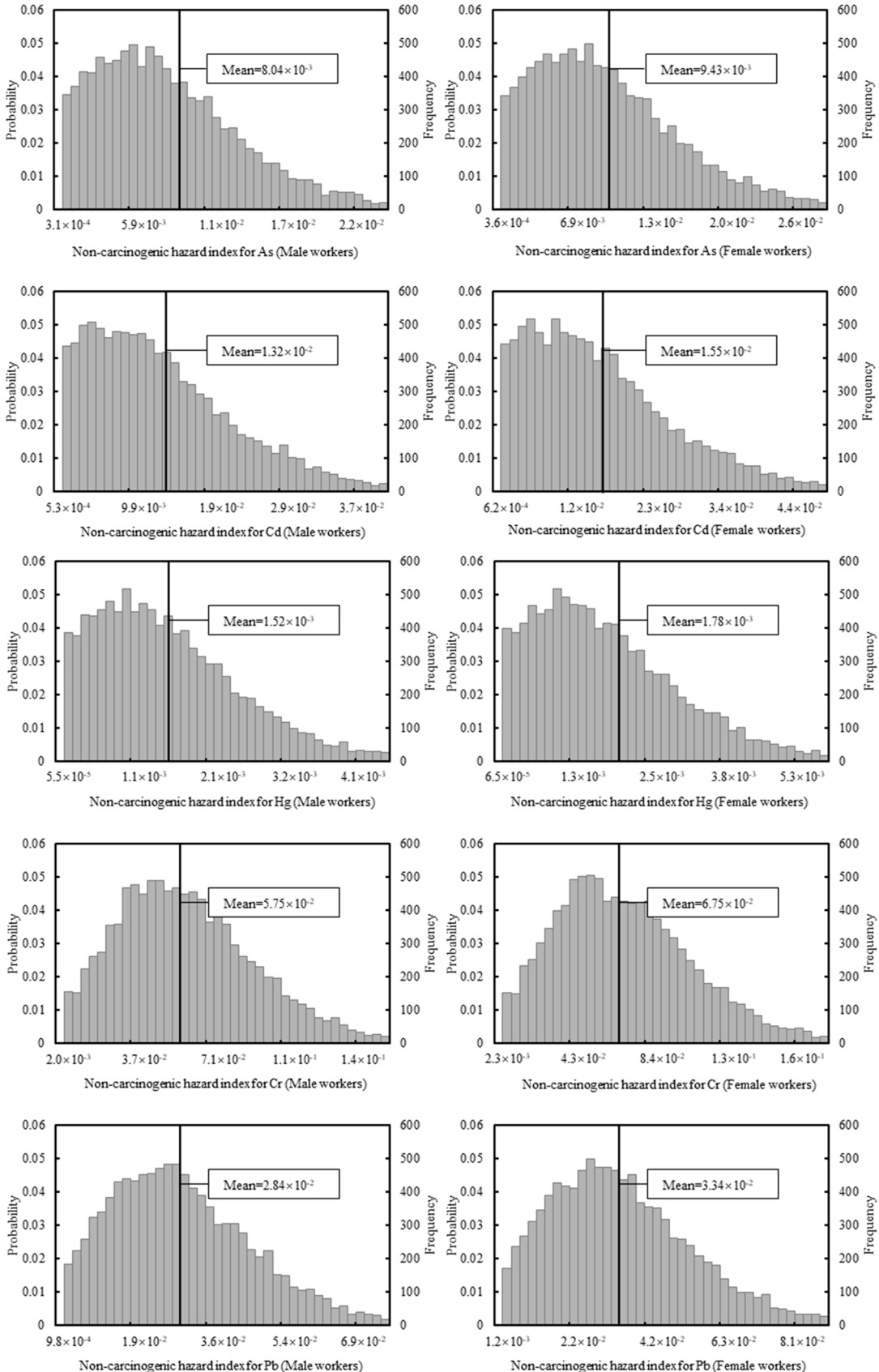

**Figure 4.** Distribution of non-carcinogenic risks posed by different heavy metals.

### 3.4.2. Carcinogenic Risk

Since the SF values of Hg for three exposure pathways were missing and the SF values of Cd, Cr, and Pb were not completed (refer to Table A2), only three exposure pathways of As, two exposure pathways of Cr (hand-to-mouth ingestion, and air inhalation), and one exposure pathway of Cd (air inhalation) and Pb (hand-to-mouth ingestion) were taken into consideration for the carcinogenic risk assessment.

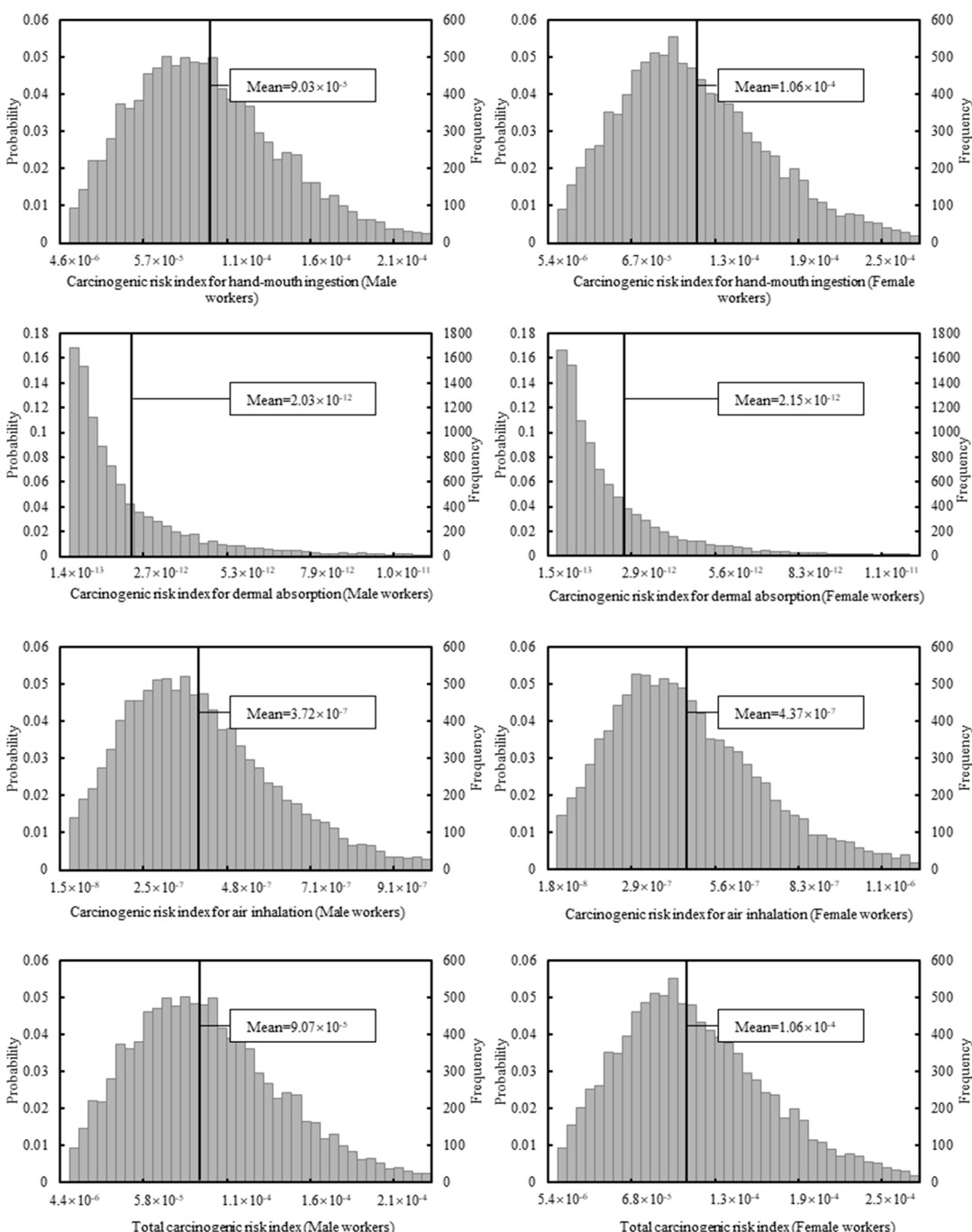

**Figure 5.** Distribution of carcinogenic risks posed through different exposure pathways.

The results in Figure 5 indicate that: (1) the mean carcinogenic risk indices of hand-to-mouth ingestion, dermal absorption and air inhalation for male workers were $9.03 \times 10^{-5}$, $2.03 \times 10^{-12}$ and $3.72 \times 10^{-7}$, respectively; (2) the mean carcinogenic risk indices of hand-to-mouth ingestion, dermal

absorption and air inhalation for female workers were $1.06 \times 10^{-4}$, $2.15 \times 10^{-12}$ and $4.37 \times 10^{-7}$, respectively; (3) the mean total carcinogenic risk indices for male and female workers were $9.07 \times 10^{-5}$ and $1.06 \times 10^{-4}$, respectively. The risk indices distribution of different exposure pathways indicated that the carcinogenic risk indices of hand-to-mouth ingestion, dermal absorption and air inhalation for male and female workers were distributed in the range of $10^{-6} \sim 10^{-4}$, $10^{-13} \sim 10^{-11}$, and $10^{-8} \sim 10^{-6}$, respectively. The risk indices of hand-to-mouth ingestion was 7 and 5 orders of magnitude higher than that of dermal absorption and air inhalation, respectively.

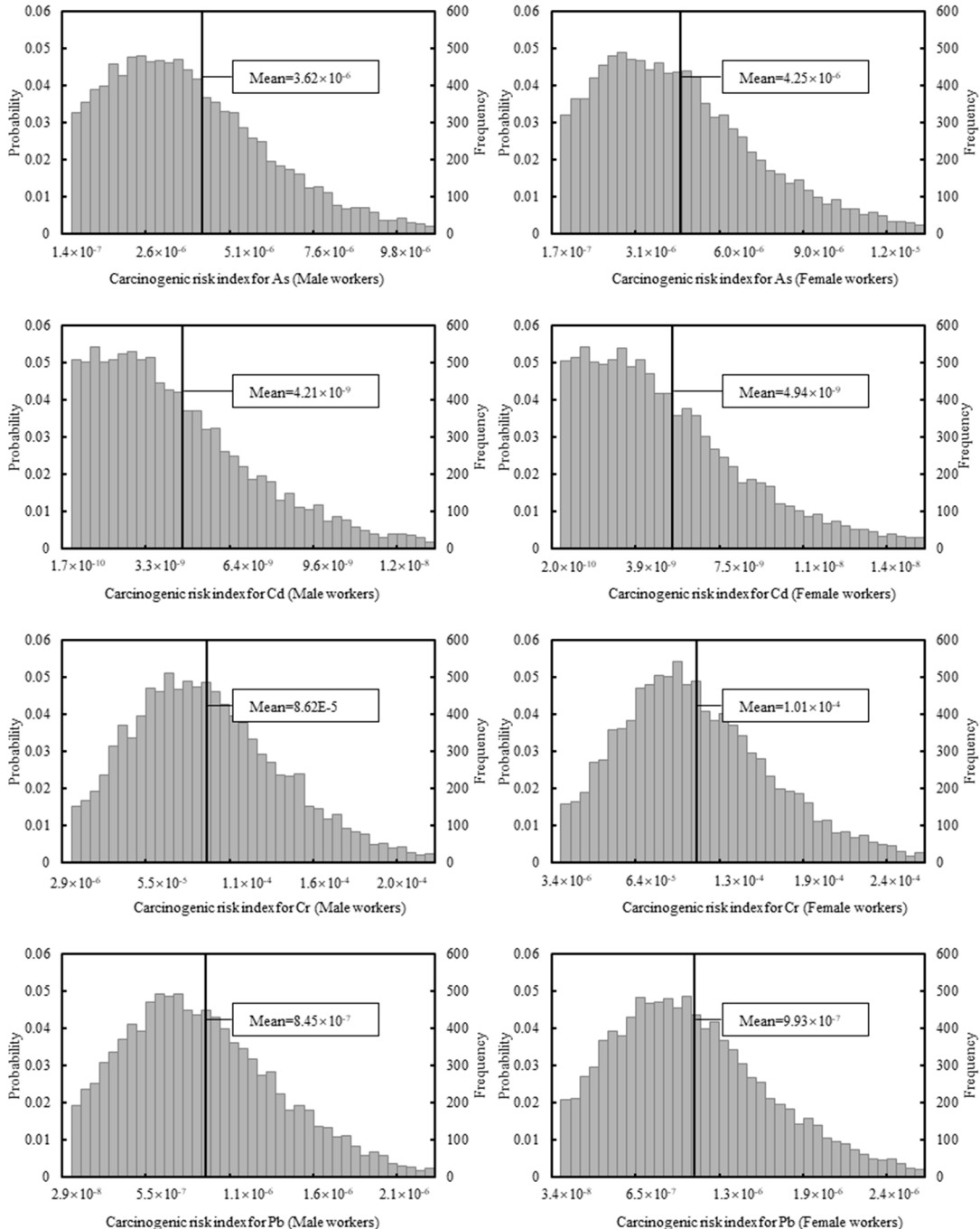

**Figure 6.** Distribution of carcinogenic risks posed by different heavy metals.

The results in Figure 6 show that: (1) the average carcinogenic risk indices of As, Cd, Cr and Pb for male workers were $3.62 \times 10^{-6}$, $4.21 \times 10^{-9}$, $8.62 \times 10^{-5}$ and $8.45 \times 10^{-7}$, respectively; (2) the

average carcinogenic risk indices of As, Cd, Cr and Pb for female workers were $4.25 \times 10^{-6}$, $4.94 \times 10^{-9}$, $1.01 \times 10^{-4}$ and $9.93 \times 10^{-7}$, respectively. The risk indices distribution of different heavy metals showed that the carcinogenic risk indices of As, Cd, Cr and Pb for male and female workers were distributed in the range of $10^{-7} \sim 10^{-5}$, $10^{-10} \sim 10^{-8}$, $10^{-6} \sim 10^{-4}$, and $10^{-8} \sim 10^{-6}$, respectively. Therefore, Cr contributed the most to the carcinogenic risk for male and female workers, following by As, Pb and Cd.

### 3.5. Sensitivity Analysis

Since the health risk assessment model involves more than 20 variables and uncertainty is inevitable during the Monte Carlo simulation, sensitivity analysis was used to determine the sensitivity factors that have important influence on health risk assessment. The Crystal Ball Software 11.1 was used for the sensitivity analysis and results were listed in Figure 7.

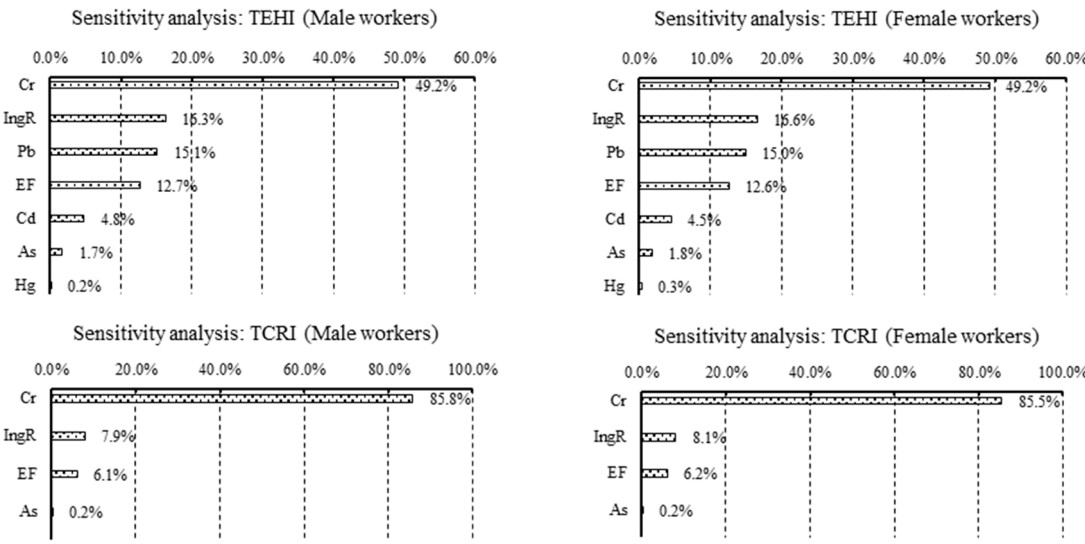

**Figure 7.** Sensitivity analysis for total exposure risk.

Figure 7 shows that the concentration of Cr in MSW was the most sensitive parameter contributed the most to TEHI and TCRI for male and female workers. The other sensitive parameters for TEHI were IngR, Pb, EF, Cd, As and Hg, accounting for (16.3%~16.6%), (15.0%~15.1%), (12.6%~12.7%), (4.5%~4.8%), (1.7%~1.8%), and (0.2%~0.3%) of the total variance, respectively. The other sensitive parameters for TCRI were InhR, EF, and As, accounting for (7.9%~8.1%), (6.1%~6.2%), and 0.2% of the total variance, respectively.

## 4. Discussion

### 4.1. Composition of MSW

The composition of MSW is different in different societies, and in previous work, the bigger factors influencing MSW composition appear to be wealth and consumption habits [37]. In the Eastern Guangdong, food waste was the largest component, followed by plastic and paper. This was consistent with results of other domestic studies and researches [6,29]. However, in the United States, paper component was the largest component, followed by food waste and textile. In France, food waste was the largest component, followed by paper and textile [37]. This difference should be caused by different consumption habits [37]. The soil component in Eastern Guangdong MSW (average mass fraction was 16.98%) was relatively higher than areas like the Pearl River Delta but lower than Guangdong rural areas [6,38]. It is speculated that this is primarily attributed to the fact that the Eastern Guangdong remains an area under development, and its economic growth and urban development are somewhere between rural areas and developed cities. As revealed by the results obtained from previous studies,

the mass fraction of soil component in the MSW conforms to a rule, that is, rural area > suburb > urban [29]. The mass fractions of other components, such as paper, glass, textile, and metal, fail to show any significant difference from the data of MSW collected from other regions in China [6,38].

### 4.2. Heavy Metal Concentrations in MSW

Compared to China's rural MSW, the mean Pb concentration in the Eastern Guangdong MSW was slightly higher than that in China's rural MSW, but the mean Cd, Hg and Cr concentrations in the Eastern Guangdong MSW were significantly lower than that in China's rural MSW [39]. Comparing to other domestic cities, the average Cr, Hg, and Pb concentrations in the Eastern Guangdong MSW were 62%, 75%, and 60%, respectively, lower than that in Guangzhou MSW, while the average Cd content in the Eastern Guangdong MSW was 50% higher than that in Guangzhou MSW [6]. The average Cd and Cr contents in the Eastern Guangdong MSW were one time higher than that in Shanghai MSW, but the average concentrations of As, Hg and Pb were almost the same as that in Shanghai MSW [7]. In Table 1, the As, Cd, Hg, Cr, and Pb variation coefficients (99%, 207%, 127%, 58%, and 67%, respectively) were greater than 50%, indicating a high discrete degree. This is mainly determined by the heterogeneous characteristics of MSW, and secondly it was also influenced by the season and climate change in the sampling areas because the sampling process covered a wide area and took a long time-interval [29]. Overall, the As, Cd, Hg, Cr, and Pb concentrations in the Eastern Guangdong MSW obey standard normal distribution.

### 4.3. Possible Source of Heavy Metals

As shown in Table 2, the Pb and Cr content in MSW was significant positively correlated at 0.01 level, which suggested that they potentially came from the same source. The As content was significant positively correlated with mass fraction of soil and the Cd content was significant positively correlated with mass fraction of food waste, indicating that the As and Cd in MSW were possibly derived from soil and food waste, respectively. The Pb content in MSW was significant positively correlated the mass fractions of glass, textile, food waste, and white plastic, which suggested that these mentioned components were Pb's potential sources. Furthermore, the significant positive correlation between and among food waste, glass, white plastic, plastic and paper implied that the generation of glass, white plastic, plastic and paper were to a certain extent related to food waste, because the glass, plastic, white plastic, and paper are usually used as packaging materials for food. In addition to these positive correlations, the As content with mass fractions of food waste, the Cd content with mass fractions of paper, the Hg and Cr contents with mass fractions of soil were negatively correlated. These negative correlations indicated that the concentrations of negatively correlated heavy metals decreased as the mass fraction of these components increased in MSW.

The factor analysis results (refer to Table 3) showed that two principal components could explain the sources of As, Cd, Hg, Cr and Pb in the Eastern Guangdong MSW. The first factor, F1, accounted for 72.1% of the total variance. Hg, Cr and Pb had high F1 loadings. Pearson correlation analysis indicated that Cr and Pb have the same source and Pb might derived from glass, textile, food waste, and white plastic. In addition, as packaging materials, the glass, plastic, paper, and white plastic have strong correlations with food waste. Thus, F1 could be explained as food waste and packaging materials, including plastic, paper, and glass, which was the main influencing factor affect heavy metal contents in the Eastern Guangdong MSW. The second factor, F2, accounted for 24.0% of the total variance. As and Cd had high F2 loadings. As previously described, As and Cd in MSW were possible respectively derived from soil and food waste. Therefore, F2 could be explained by soil components being mixed into food waste. As the third major component in the MSW, mass fraction of soil component in the MSW has reached up to 16.98%, being another major factor affecting heavy metal contents in the Eastern Guangdong MSW.

### 4.4. The Health Risks Posed by Heavy Metals in MSW

The cumulative risk probability curve reflects the probability distribution of non-carcinogenic diseases or cancers in MSW workers exposed to heavy metals. Cumulative probability curves of the TEHI and the TCRI for MSW workers were listed in Figure 8a,b, respectively.

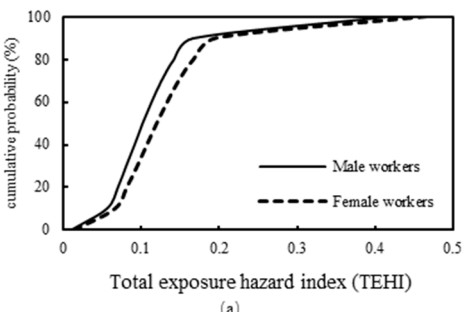 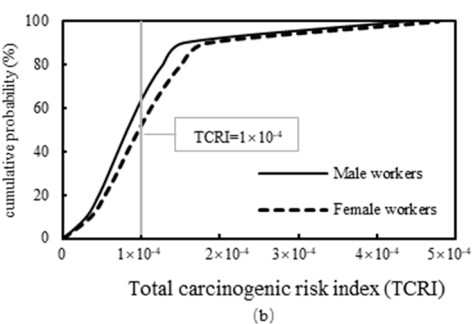

(a)                                                (b)

**Figure 8.** Cumulative probability curves: (**a**) cumulative probability of the TEHI for MSW workers; (**b**) cumulative probability of the TCRI for MSW workers.

As shown in Figure 8, the cumulative probability curves of the non-carcinogenic and carcinogenic risk for female workers were always below the male workers' curves. This indicated that, under the same exposure environment, the non-carcinogenic hazard and carcinogenic risk for male workers were both slightly lower than that of female workers.

According to the US Environmental Protection Agency handbook guidance [25], no non-carcinogenic risks are posed to the exposed individuals when the HI < 1 and the exposed individuals are under the risks of non-carcinogenic diseases when HI > 1, with the risk probability increasing as HI increases. The Figure 8a showed that the TEHI for male workers and female workers were both below 0.5 when the cumulative probability was reached 100%, indicating that the non-carcinogenic risks for MSW workers were negligible.

According to previous studies [34], TCRI > $1 \times 10^{-4}$ was considered unacceptable, TCRI < $1 \times 10^{-6}$ was considered safe, and TCRI between $1 \times 10^{-6}$ and $1 \times 10^{-4}$ were considered acceptable or tolerable. The Figure 8b showed that the cumulative risk probability of male workers and female workers were respectively 65% and 55% when the TCRI was equal to the absolute safety threshold ($1 \times 10^{-4}$). This implies that the carcinogenic risk probabilities for the male workers and female workers in the Eastern Guangdong were 35% and 45%, respectively.

Risk indices of different exposure pathways indicated that hand-to-mouth ingestion was the major exposure way of heavy metals leading to non-carcinogenic and carcinogenic risks for male and female workers. As previously explained, landfill was the ordinary disposal of MSW in the Eastern Guangdong. Unlike waste incineration, this disposal way will not release large amounts of heavy metals into the air and surrounding environment. Furthermore, few skin areas were directly exposed to the MSW. Thus, dermal absorption and air inhalation didn't contribute much to the non-carcinogenic and carcinogenic risk indices. Risk indices of different heavy metals indicated that Cr contributed the most to the non-carcinogenic and carcinogenic risks. Besides, result of sensitivity analysis also demonstrated that IngR, EF and heavy metals concentrations (especially for Cr) were the key parameters affecting health risks. Taken together, in order to reduce the health risks of MSW workers, first of all, effective countermeasures must be taken to disrupt the heavy metals exposure pathways (especially for the hand-to-mouth ingestion) to human body, and then to reduce the exposure time and heavy metal concentrations in MSW.

## 5. Conclusions

The MSW from Eastern Guangdong was mainly comprised of soil, glass, metal, paper, plastic, grass, food waste, and white plastic. MSW composition is consistent with a majority of other domestic

cities. Nevertheless, the top three components of MSW were significantly different compared to the United States and France. This may be due to different consumption habits in different societies. The mass percentage of soil component in the Eastern Guangdong MSW was noticeably higher than that in the MSW from the Pearl River Delta but lower than that in MSW from the rural areas in Guangdong. It is probably caused by the difference in economic growth and urban development.

The contents of As, Cd, Hg, Cr, and Pb in the Eastern Guangdong MSW conform to the standard normal distribution, and their concentrations were (0.76 ± 0.75), (2.14 ± 4.44), (0.11 ± 0.14), (55.42 ± 31.88) and (30.67 ± 20.58) mg/kg, respectively. There are two major influencing factors that have impact on the content of heavy metal in the Eastern Guangdong MSW. One is food waste and packaging materials, including plastic, paper, and glass, and the other is soil component mixed into food waste. Hg, Cr and Pb are possible to be derived from glass, textile, food waste, and white plastic, while As and Cd in MSW were possibly derived from soil and food waste, respectively.

If MSW workers work continuously for 30 years, the non-carcinogenic risk of exposure to heavy metal can be neglected. However, the probability of carcinogenic risk faced by the male workers and female workers in the Eastern Guangdong was 35% and 45%, respectively. The Cr content contributed the most to the carcinogenic risks, and the most dangerous exposure pathway was hand-to-mouth ingestion. Effective countermeasures, such as exposure pathways disruption, exposure time and heavy metal contents reduction, must be taken to mitigate the health risks for MSW workers.

**Author Contributions:** All authors read and approved the manuscript. All authors contributed to this work, discussed the results and implications and commented on the manuscript at all stages. Z.T. gave valuable advice on the establishment of the framework, as well as the design process. M.L. and L.Y. collected samples and did chemical test. H.G. and T.O. discussed the main idea behind the work and reviewed and revised the manuscript. H.Y. enhanced the quality of the manuscript at all stages. M.L. developed the revised manuscript.

**Funding:** This research was funded by the Natural Science Foundation of Guangdong Province, grant number 2018A030310084 and was funded by the National Natural Science Foundation of China, grant number 21606228.

**Acknowledgments:** The authors are grateful to Ding Xinyang from Boston College for language polishing.

**Conflicts of Interest:** The authors declare no conflict of interest.

**Appendix A**

**Table A1.** Parameters and input assumptions used in the health risk assessment.

| Parameter | Units | Type | Distribution |
|---|---|---|---|
| CMSW | (mg/kg) | Normal | Hg: $\mu = 0.1$; $\sigma = 0.14$<br>Cd: $\mu = 2.14$; $\sigma = 4.44$<br>Pb: $\mu = 30.67$; $\sigma = 20.58$<br>Cr: $\mu = 55.42$; $\sigma = 31.88$<br>As: $\mu = 0.61$; $\sigma = 0.67$ |
| IngR | kg/day | Log-normal | $\mu = 24 \times 10^{-5}$; $\sigma = 4 \times 10^{-5}$ |
| InhR | m$^3$/day | Log-normal | $\mu = 16.57$; $\sigma = 4.05$ |
| EF | day/year | Triangular | c = 345; a = 180; b = 365 |
| ED | year | Point | 30 |
| W | kg | Log-normal<br>Log-normal | $\mu = 67.55$; $\sigma = 8.72$ [1]<br>$\mu = 57.59$; $\sigma = 8.03$ [2] |
| SA | m$^2$ | Triangular | c = 0.169; c = 0.085; b = 0.422 [1]<br>c = 0.153; a = 0.076; b = 0.382 [2] |
| AF | mg/(cm$^2$·d) | Log-normal | $\mu = 0.49$; $\sigma = 0.54$ |
| ABS | unitless | Point | 0.14 (Cd), 0.04 (Cr), 0.03 (As), 0.05 (Hg), 0.006 (Pb) |

[1] Male workers; [2] Female workers.

**Table A2.** Values of RfD (mg/(kg·d) and SF (kg·d)/mg) for five heavy metals.

| | As | Cd | Hg | Cr | Pb |
|---|---|---|---|---|---|
| RfD for hand-mouth ingestion | $3.00 \times 10^{-4}$ | $1.00 \times 10^{-3}$ | $3.00 \times 10^{-4}$ | $3.00 \times 10^{-3}$ | $3.50 \times 10^{-3}$ |
| RfD for dermal absorption | $1.23 \times 10^{-4}$ | $1.00 \times 10^{-5}$ | $2.10 \times 10^{-5}$ | $6.00 \times 10^{-5}$ | $5.25 \times 10^{-4}$ |
| RfD for inhalation | – | $1.00 \times 10^{-5}$ | $8.57 \times 10^{-5}$ | $2.86 \times 10^{-5}$ | – |
| SF for ingestion | 1.50 | – | – | $5.00 \times 10^{-1}$ | $8.50 \times 10^{-3}$ |
| SF for dermal absorption | 3.66 | – | – | – | – |
| SF for inhalation | $1.51 \times 10^{1}$ | 6.30 | – | $4.20 \times 10^{1}$ | – |

Note: – means data missing.

**Table A3.** Eigenvalue and accumulating contribution rate.

| Component | Initial Eigenvalues | | | Extraction Sums of Squared Loadings | | | Rotation Sums of Squared Loadings | | |
|---|---|---|---|---|---|---|---|---|---|
| | Total | Variance % | Cumulative % | Total | Variance % | Cumulative % | Total | Variance % | Cumulative % |
| PC1 | 3.61 | 72.10 | 72.10 | 3.61 | 72.10 | 72.10 | 3.56 | 71.15 | 71.15 |
| PC2 | 1.20 | 24.02 | 96.13 | 1.20 | 24.02 | 96.13 | 1.25 | 24.97 | 96.13 |
| PC3 | 0.12 | 2.35 | 98.47 | | | | | | |
| PC4 | 0.06 | 1.30 | 99.77 | | | | | | |
| PC5 | 0.01 | 0.23 | 100.00 | | | | | | |

Extraction method: Principal component analysis.

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
