# Peer review of "Source Apportionment and Health Risk Assessment of Heavy Metals in Eastern Guangdong Municipal Solid Waste"

_applsci, doi:10.3390/app9224755_

Round 1

Reviewer 1 Report

The scientific problem and logic in this study is clear. The Pearson correlation and principal component analysis (PCA) are relatively mature research methods used for heavy metal source apportionment. The innovation of the article is Monte Carlo method combined with the US EPA Model, which ensured the scientific and reasonable for the results of risk assessment. The work is interesting and inspiring to the field of environmental protection and public health. I therefore recommend this paper to be published. However, there are few issues, it is better to revise before submitting the final version. 

In the line 177: the font size of ordinate title in Figure 2 is too big and do not need to be bold. The experimental works (e.g., Green Energy and Environment. 2019, Doi.: 10.1016/j.gee.2019.05.002) on the removal of Pd(II) and Cd(II), it is much better to mention.

Author Response

Dear professor:

Thank you for your very valuable and constructive comments, we have carefully revised the manuscript point by point.

Point 1: In the line 177: the font size of ordinate title in Figure 2 is too big and do not need to be bold. The experimental works (e.g., Green Energy and Environment. 2019, Doi.: 10.1016/j.gee.2019.05.002) on the removal of Pd(II) and Cd(II), it is much better to mention. 

Response 1: The mistake in Figure 2 has been revised. The mentioned article has been carefully studied and added in the references.

Reviewer 2 Report

The work concerns an interesting topic and is carried out in an overall satisfactory manner. However, some parts are unclear and should be expanded:

2.2 Sample Preparation and Testing:

 In this section it is not clear that the first part of the MSW sample analysis process concerns the determination of the mass percentage of MSW to the components soil, glass, metal, paper, plastic, textile, grass, food waste, and white plastic like shown in fig 2 of section 3.1.

2.3. Statistical Analysis

This section should be expanded, it should be explained that the statistical techniques applied to the components of MSW and to metals are used to determine the source apportionment, one of the objectives of the work.

The result of Pearson correlation analysis and the factor analysis results should be reported in section 3 and not in section 4.

The conclusions should be more extensive.

Author Response

Dear professor:

Thank you for your very valuable and constructive comments, we have carefully revised the manuscript point by point.

Point 1:In the section of “2.2 Sample Preparation and Testing” it is not clear that the first part of the MSW sample analysis process concerns the determination of the mass percentage of MSW to the components soil, glass, metal, paper, plastic, textile, grass, food waste, and white plastic like shown in fig 2 of section 3.1.

Response 1: This section has been significantly improved through English proofreading services supported by professional organization (refer to Certificate of editing in attachment). We believe the MSW sample analysis process is clearer.

Point 2: The section of “2.3. Statistical Analysis” should be expanded, it should be explained that the statistical techniques applied to the components of MSW and to metals are used to determine the source apportionment, one of the objectives of the work.

Response 2: This section has been expanded according to your comments.

Point 3: The result of Pearson correlation analysis and the factor analysis results should be reported in section 3 and not in section 4.

Response 3: We have placed the result of Pearson correlation analysis in section 3.3.

Point 4: The conclusions should be more extensive.

Response 4: We have expanded this section.

Reviewer 3 Report

The manuscript requires proofreading and improvement. Some sentences do not read well. Writing style requires serious improvement.

As for the conclusion section, the statements should be linked to their findings and/or the current literature.

The materials and methods section requires improvement. Again, their writing style requires serious improvement.

Overall, it is an interesting analysis for the readers.  

Author Response

Dear professor:

Thank you for your very valuable and constructive comments, we have carefully revised the manuscript point by point.

Point 1: The manuscript requires proofreading and improvement. Some sentences do not read well. Writing style requires serious improvement.

Response 1: The manuscript has been significantly improved through English proofreading services supported by professional organization (refer to Certificate of editing in attachment).

Point 2: As for the conclusion section, the statements should be linked to their findings and/or the current literature.

Response 2: We have improved the conclusion section according to your comments.

Point 3: The materials and methods section requires improvement. Again, their writing style

Response 3: This section has been significantly improved through English proofreading services supported by professional organization(refer to Certificate of editing in attachment).
